# Breed Varieties of Pigs for Disease Resistance and Susceptibility to Seneca Valley Virus Infection

**DOI:** 10.3390/ijms26178746

**Published:** 2025-09-08

**Authors:** Wentao Wang, Fengze Han, Xinmiao He, Shihan Zhao, Ziluo Zou, Ming Tian, Heshu Chen, Saihui Wu, Yan Sun, Yaokun Jiang, Meiqin Sun, Libing Zhang, Kunzhi Yu, Yao Wang, Yaguang Tian, Xinpeng Jiang, Di Liu

**Affiliations:** 1Key Laboratory of Combining Farming and Animal Husbandry, Ministry of Agriculture, Animal Husbandry Research Institute, Heilongjiang Academy of Agricultural Sciences, No. 368 Xuefu Road, Harbin 150086, China; 2College of Animal Science and Technology, Northeast Agricultural University, Harbin 150030, China; 3School of Life Sciences, Peking University, Beijing 100871, China

**Keywords:** Min pig, seneca valley virus (SVV), disease resistance, macrophages, pig breeds

## Abstract

In recent years, outbreaks of Seneca Valley virus (SVV) in pig farms have raised concerns about disease resistance in different pig breeds. Min pigs are an excellent local pig breed in China, but the breed’s strong disease resistance mechanism has not been clearly investigated. In this study, Min pigs and Landrace pigs were challenged with SVV, and the differences in pathogenicity between SVV-infected Min and Landrace pigs were evaluated in terms of production performance, survival rate, immune cell activity, pathological changes, viral titer, and cytokine expression. The results show that the mortality rate in Min pigs was significantly lower than that in Landrace pigs without substantial weight loss. The copy number of SVV RNA in the intestinal mucosa of Min pigs was lower than that in Landrace pigs. Additionally, the IgA and IgG titers in Min pigs were higher than those in Landrace pigs. Both the absolute number and percentage of M1 macrophages were elevated in Min pigs relative to Landrace pigs. This study is the first to identify differences in macrophage activity between SVV-infected Min pigs and Landrace pigs. The presented results indicate the potential research value of studying innate immunity differences in disease resistance between pig breeds.

## 1. Introduction

In China, there are more pig breeds than in any other country in the world. There are 118 indigenous pig breeds in China according to the Domestic Animal Diversity in the Animal Genetic Resources in China [1]. Most pig breeds have famous genetic characteristics, such as high meat quality and strong resistance to crude feed, general diseases, and cold. Min pigs are one of the indigenous pig breeds in the northeast of China, which is known to have outstanding disease resistance. However, different disease models have various host sensitivities in different pig breeds, as different genotypes of same the virus can generate different symptoms and phenotypes in the same pig breed. Dingyuan pigs have been shown to be the most susceptible to PRRSV (Porcine Reproductive and Respiratory Syndrome virus) infection, whereas Jiangquhai pigs are the least susceptible to PRRSV infection [2]. The local pig breeds (Tongcheng, Meishan, and Qingping) exhibited stronger resistance to PRRSV compared to imported breeds (Landrace, Yorkshire, and Duroc) [3]. Additionally, Dalian-type pigs demonstrated higher resistance not only to PRRSV but also to Porcine Circovirus Type 2 (PCV2) [4]. Two telomere-to-telomere pig genome assemblies and a pan-genome were analyzed in the Min pig and Rongchang pig, which provides insights into core genes involved in essential processes such as transcriptional regulation and immune responses [3]. The composition of the gut microbiota in adult Min pigs also exhibits better anti-inflammatory effects compared to that in Yorkshire pigs in a DSS-induced colitis model [4]. Pig breeds may offer advantages in terms of disease susceptibility loci, and such breed-specific variations suggest opportunities for breeding disease-resistant or disease-tolerant animals.

Seneca Valley Virus (SVV), or Seneca virus A(SVA), is a single-stranded, positive-sense RNA virus that belongs to the genus Seneca virus within the family Picornaviridae and has been increasingly reported in swine breeding farms. It can lead to neonatal mortality due to swine idiopathic vesicular disease outbreaks, which can be accompanied by lethargy, diarrhea, and neurologic signs in neonates [5,6]. Experimental SVV inoculation has caused vesicular disease in pigs, particularly from the weaning to finishing stages [7]. Most importantly, the clinical positive results in older pigs and high mortality rates in piglets last for approximately 2 to 3 weeks as the virus passes through the affected herd. Previous study found that Min pigs have better disease resistance against Porcine epidemic diarrhea virus (PEDV) infection compared to Landrace pigs [8]. Both PEDV and SVV share similar pathogenic characteristics, as they both cause diarrhea and high mortality rates in neonatal piglets. However, the transmission routes and mechanisms of SVV susceptibility have not yet been elucidated. Moreover, to date, no research has been conducted on the correlation between the native pig breeds of China and their susceptibility or resistance to SVV.

In this study, we investigated the differences between Min and Landrace breed piglets in terms of pathogenicity and the immunoregulatory responses to SVV infection. This study focused on Min piglets having stronger resistance against SVV infection than Landrace piglets in terms of pathological injury, viral replication, and immune cell activity. The aim was to evaluate pathogenicity in Min piglets and to discover why their viral replication level was lower than that in Landrace piglets.

## 2. Results

### 2.1. Production Performance and Survival Rate

Once suckling piglets were challenged with SVV, the effects of diarrhea on their growth and development were observed in Min pigs and Landrace pigs. The growth performance results of live piglets indicated that there was a significant difference in the development of weight, which was the main difference between Min pigs and Landrace pigs in Figure 1A. The birth and weaning weights of Min piglets were lower than those of Landrace piglets. Certainly, SVV-infected Min piglets (SVV-Min) and Landrace piglets (SVV-Landrace) were lighter than the control groups (CON) with the infection of SVV throughout the whole weaning period. However, the weight of Landrace piglets with the SVV infection sharply decreased compared with the control group. And the weight of the Min piglets in control group did not significantly decrease compared with Min piglets after infection SVV.

There is an ongoing search for different breeds of piglets with resistance against SVV infection that can confer this trait to the host. Thus, some piglets were intranasally challenged with a dose of SVV (refer to Figure 1B). In the Landrace pig infecting group, the first mortality was observed on Day 3 post-infection, with approximately 50% surviving after the infection. We evaluated the disease resistance in Min pigs exhibiting sickness on similar days, of which only 10% piglets died due to infection. The surviving piglets began to drink milk and completely recovered. We chose Day 10 as the terminal point to assess the significant difference between groups. No piglet died in the control groups of Min pigs and Landrace pigs. These results suggest that breed characteristics increased host resistance against SVV infection. This prompted us to investigate the underlying mechanisms associated with breed-related disease resistance.

### 2.2. SVV RNA Copies Number in Two Piglet Breeds

Mouth and fecal swabs were collected to detect the mRNA of SVV for eight days after infection. The viral titers of SVV were tested via RT-qPCR, and the results indicated the values of the Ct (cycle threshold) in (Figure 2A,B). No SVV replication was observed in the Min or Landrace piglets in the control groups, and the Ct value indicated that there was no SVV infection in these two groups. The number of mRNA copies in the feces indicated that SVV significantly replicated in the Landrace piglets, at a level higher than that in Min piglets. The highest numbers of mRNA copies were observed on the fifth day after SVV infection in mouth, and that in Min piglets was lower than in Landrace piglets. The fecal swab results also showed the same trends in terms of the mRNA copies, and the fifth and sixth days also exhibited the highest copy numbers of SVV mRNA in the infected Landrace piglet group. However, Min piglets did not show significant replication following SVV infection, which indicates that the SVV mainly replicated in the intestinal tract of the Landrace piglets.

### 2.3. Intestinal Mucosal-Specific IgA and IgG

The disease resistance in different piglet breeds was evaluated. The production of specific IgA and IgG in Min piglets against SVV was better than that of Landrace piglets, reflecting the specific immune response. We detected a difference in the antibody titers of SVV in different intestinal segments from Min piglets and Landrace piglets. After the challenge with SVV, the IgA titers (Figure 3A) in intestinal mucosa and IgG (Figure 3B) in Min and Landrace pigs were significantly higher than those in the control groups. This indicates that SVV could stimulate the specific immune response in both piglet breeds. Moreover, the IgA titer in the intestinal mucosa of Min piglets was much higher than that of infected Landrace piglets. Moreover, the IgA titer of the jejunum mucosa from Min pigs infected with SVV was up to 200 doubling dilutions in the small intestine, meaning that it could effectively neutralize the SVV infection. In addition, the IgA titer of the intestinal mucosa of Landrace pigs infected with SVV was about 50 doubling dilutions in the duodenum. After the challenge with SVV, the anti-SVV IgG titer was similar to the IgA titer. The anti-SVV IgG titer in Min pigs infected with SVV was much higher than that in Landrace pigs. These results indicate that the disease-resistance to SVV of Min pigs is higher than that of Landrace pigs, in terms of the specific immune response.

### 2.4. Min Pigs Inhibit SVV Infection with Immunofluorescence Analysis

We verified the distribution of viral antigens in the intestine with fluorescence and divided them into four groups for study: negative control groups for Min piglets and Landrace piglets and infected groups for both Min piglets and Landrace piglets (Figure 4). Most importantly, Landrace piglets infected with SVV exhibited more severe pathological damage to the intestinal tissue compared with Min piglets infected with SVV. Specifically, Landrace piglets infected with SVV showed extensive detachment and necrosis of intestinal epithelial cells, whereas Min piglets infected with SVV displayed relatively intact intestinal tissue morphology. No green fluorescence was observed in either of the negative control groups. The results for the infected groups indicated that both Min and Landrace piglets were infected with SVV. However, under observation with a fluorescence microscope, the fluorescence quantity in the Min piglet group decreased, when compared with that in the Landrace piglet group. This shows that Min pigs, with the innate breeding characteristic of disease resistance, inhibited SVV RNA replication and production in the intestine. The following figure shows representative sections derived from each group.

### 2.5. Activated Macrophage Inhibiting SVV in Min Pigs

In order to validate the existing phagocyte cells, microscopic analysis was performed to assess the characteristics of macrophages and dendritic cell activity from porcine intestinal tissue sections from Min and Landrace piglets, which were marked with F4/80 and CD11b, respectively. After diarrhea was observed in the infected piglets, three piglets from each group were randomly selected for necropsy. Both phagocyte cells for macrophages and dendritic cells were analyzed in Min and Landrace piglets. The control groups of these two breeds of pigs showed no significant differences in the dendritic cells (Figure 5B,D). Additionally, there were no noticeable intestine lesions observed among the two groups. However, the number of macrophages in Min piglets was higher than in Landrace piglets, which indicates that the former has stronger innate immunity (Figure 5A,C). Through immunofluorescence analysis (IFA), the SVV antigen was found to be enhanced in the cytoplasm of villous enterocytes in both infected groups. SVV infection showed the same trend as the macrophages, which were more numerous in Min than in Landrace piglets.

### 2.6. M1 Polarization from Min Pigs Against SVV Infection

To further characterize the porcine macrophages and their M1/M2 polarization, splenic lymphocytes from Min and Landrace piglets were studied with flow cytometry in (Figure 6A,B). We first tested whether there was a difference in the macrophage subtypes between Min and Landrace piglets, which could affect SVV infectivity. The M1 and M2 macrophages results from both piglet groups indicated that Min pigs had the M1 breeding characteristic, which plays a significant role in defense against pathogenic microorganism infection (Figure 6A). Most importantly, the balance of M1 and M2 macrophages in Min piglets differed from that in Landrace piglets. Moreover, M1/M2 macrophages were unbalanced in uninfected Min piglets, and the level of M1 macrophages in the Min piglets was higher than in the other breeds. There was significant difference between Min piglets and Landrace piglets in the M1 macrophage polarization (Figure 6B). Notably, the infected groups showed similar results, M1 macrophage production in Min piglets was stimulated by the SVV infection, and the number and percentage of splenic lymphocytes were also higher than in the Landrace piglets.

### 2.7. Induction of M1 Polarization Associated with SVV Is Mediated Through the Activation of IFN-λ and IL-6

To identify how M1 polarization is induced in Min pigs compared to that in Landrace pigs with and without SVV infection, the relative protein concentrations of signaling molecules in spleen tissues were studied via Western blot. Firstly, we sought to investigate the mechanisms by which M1 polarization was induced, which was stronger in Min than in Landrace piglets. From the results, we found that the expression levels of IFN-λ (Figure 7C) and IL-6 (Figure 7E) in the Min group were higher than in the Landrace group (Figure 7C). Both of these are M1 macrophage markers that are linked to inflammatory processes. However, the type I IFN of IFN-α (Figure 7A) and TGF-β (Figure 7D) did not show high expression in these two breeds. Both IFN-λ and IL-6 can stimulate M1 differentiation. The Min piglets exhibited the breed characteristic of M1 macrophage activity in the immune system. Stronger M1 macrophage activity also plays an important role against SVV infection in both pig breeds. However, Min piglets presented stronger phagocyte activity due to the M1 macrophages.

## 3. Discussion

At the clinical level, the observation of vesicular lesions could affect the breastfeeding and feeding of pigs with SVV infection. Our investigations indicated that production performance and weight decreased in SVV-infected Landrace piglets. Most importantly, neonatal pre-weaning mortality was higher in Landrace piglets compared to SVV-infected Min piglets in this study, which exhibited similar pathological damage characterized by vesicular lesions and mild-to-moderate neonatal diarrhea associated with decreased weight [9]. However, none of these results explained why the pigs tested positive for SVV via PCR, as no gross anatomic or histologic lesions were observed in neonatal pigs [10]. A previous study reported information regarding how experimental SVV infection affects gestating sows and neonates, indicating that SVV is a causative virus for the abovementioned clinical signs in neonates [11]. The neonatal mortality rate of SVV infection in this study was similar to that in a previous study in 4- to 7-day-old piglets [9].

In the intestines of Min piglets, mRNA of SVV was not detected in either of the infected or control groups, and the copy numbers of SVV RNA from Min piglets was lower than that in Landrace piglets regarding the mouth swabs. VP1 has been reported to elicit an antibody-neutralizing response, which allows us to distinguish it from other specific immunity mechanisms [12,13]. The recombinant expression of SVV-VP1 protein was used to test the specific IgG and IgA in sera and different intestinal segments with the ELISA method. Positive serum samples from piglets not only demonstrated active immune response but also presented detectable SVA-VP1 IgG maternal antibodies during the first month. However, intestinal segment samples did not contain the specific IgA related to maternal antibodies, but the IgA index demonstrated an active immune response from the mucosal immune system. Our results revealed that Min piglets have stronger mucosal immunity than Landrace piglets, which may explain why the former have breed characteristics that enable them to resist SVV infection via mucosal immunity. However, none of these data indicate the clear mechanism of disease resistance in Min pigs. There is no available information regarding the immune response relating to disease resistance in Min pigs throughout the duration of SVV infection under either natural or experimental conditions.

Antiviral innate immunity primarily depends on phagocytes, such as DCs and macrophages, which play important roles in defending against viral infection in the host. However, we did not establish such a relationship or observe differences in their innate immunity against SVV infection. Studies on viral and bacterial infections have shown that variations in disease susceptibility or pathology within breeds are correlated with differences in macrophage activation [14,15,16]. This approach provides a convenient basis for the analysis of genetic variations in pigs’ pathogenetic interactions through the use of in vitro challenge models. Here, the results of immunofluorescence and flow cytometry observed that the macrophages from Min piglets had stronger differentiation and development abilities than those from Landrace piglets, and M1 was the key role in Min pig stronger macrophage activity than Landrace pig, which is a phenotype likely influenced by breed-specific genetic background. A previous study found substantial interindividual variation between pigs within breeds, mostly affecting the expression of genes involved in immune responses [17,18]. Landrace pigs expressed substantially lower IL-8Rβ levels than other breeds, such as Hampshire, Pietrain, and Duroc, and Landrace macrophages were more resistant to PRRS virus replication and released large amounts of inflammatory factors [19,20].

There was much greater variation between individuals within breeds, exhibited through differences based on the relative differentiation and development of macrophages. We examined the differences in the expression of cellular markers regarding macrophages between the non-infectious and infectious groups of the two breeds. Amongst the differences, higher expression levels of IFN-λ and IL-6 were apparent, both of which can stimulate macrophage differentiation [21,22]. In addition, TGF-β in the Min piglet control group had a lower expression level than in the Landrace piglet control group; notably, TGF-β can inhibit macrophages from undergoing M1 to M2 development. Differences between cytokines have been found in previous studies: stimulation with TGF-β, induced the up-regulation of M2 macrophages [23], which explains the M1 polarization observed in Min pigs. Conversely, while TGF-β is important in terminating proinflammatory and antiviral immune responses, the M1 immune response plays a role at the beginning of SVV infection [24]. A previous study supported that TGF-β plays a key role in DENV-2 replication in macrophages, and suggested that targeting TGF-β may represent an alternative therapeutic strategy to be explored in the context of dengue infection [25,26,27,28,29,30]. All these results indicate that individual pigs vary most markedly in their expression of immune-associated inducible proteins, and there are major breed-dependent variations in IFN-λ and IL-6 cell signaling, thus influencing the stimulation of M1 differentiation in Min pigs. These findings, based on SVV infection, suggest that Min pigs have stronger M1 macrophage activity and higher IFN-λ expression for immune-associated traits can benefit their disease resistance compared with Landrace pig.

## 4. Materials and Methods

### 4.1. Ethics Statement

This study adhered to the animal welfare guidelines of the World Organization for Animal Health. All clinical animal samples used in this study were approved by the Committee on Ethics from the Animal Science and Technology College of Northeast Agricultural University for routine testing. The animal health code was NEAUEC20250257.

### 4.2. Cell Line and Virus

BHK21 cells were used for the proliferation of SVV. SVV was cultured in Baby hamster kidney 21 (BHK-21) cells in Dulbecco’s modified Eagle’s medium (DMEM) containing 10% fetal bovine serum (FBS), penicillin, and streptomycin at 37 °C with 5% CO_2_. Viral titers were tested using a 50% tissue culture infective dose (TCID50), and the per milliliter amount was calculated using standard statistical methods. The viral units used in this study were 108.5 PFU mL^−1^.

### 4.3. Experimental Infection and Experimental Design

Min pigs and Landrace pigs were obtained from the Key Laboratory of Combining Farming and Animal Husbandry, Ministry of Agriculture, Animal Husbandry Research Institute, Heilongjiang Academy of Agricultural Sciences. The SVV-CH-09-2018 strain was used for the challenge test. Min and Landrace weaning piglets were used in our study. SVV, Foot-and-Mouth Disease Virus (FMDV), Swine vesicular disease virus (SVDV), Vesicular Stomatitis Virus (VSV), and pseudorabies were not detected with the corresponding ELISA antibody kits or RT-PCR or PCR methods. The Min and Landrace weaning piglets were divided into four groups: the Min control group, the Min SVV-infected group, the Landrace control group, and the Landrace SVV-infected group. Each experimental group consisted of twenty piglets, which were housed in separate rooms to ensure consistent and standardized infecting experiments. Both infection groups underwent oral gavage with the SVV-CH-09-2018 strain, 2 mL (1 × 10^8.5^ TCID_50_/mL), while the negative control (CON) groups of Min and Landrace piglets were inoculated with DMEM at the third day of birth. All groups of piglets were housed under same conditions and kept in separate rooms, with strict biosafety protocols implemented to prevent cross-infection. Following infection, clinical symptoms were monitored, and fecal and mouse swab samples were collected daily for nine days. Each group randomly chose five piglets for sampling of feces and sera swabs. At 10 days post challenge, three piglets were euthanized for pathological examination. The gut, spleen, inguinal lymph nodes, and other organs were taken for histopathological observation. The day after inoculation was considered as day 0, and the piglets’ weight changes, survival rate, and diarrhea were observed over the following 10 days.

### 4.4. qRT-PCR Analysis

Viral loads in fecal and oral samples with the fecal and oral swabs collected from all piglets were quantified using the SVV-specific qPCR method in our study [31]. The TaqMan real-time RT-PCR was used to detect the specific mRNA of SVV titers in pig samples and the viral load as described above [32] The extracted RNA was subjected to real-time qRT-PCR using the One-Step SYBR^®^ qRT-PCR kit (Takara Corporation, Shiga Prefecture, Japan). Following reverse transcription of viral RNA at 45 °C for 20 min, the resulting cDNA was used as a template for real-time PCR amplification. A standard curve was generated by plotting the threshold values against the serially diluted plasmid DNA encoding the SVV VP1 gene fragment. The primer of VP1 gene fragment were SVV-F AACCGGCTGT GTTTGCTAGAG and SVV-R GAACTCGCAGACCACACCAA; the probes was SVV-P 6/FAM-TCGAGAAGCTGCAATCTG/MGB-NFQ [33].

### 4.5. Histopathology and IFA Assessment

The intestinal tissues from Min and Landrace pigs were fixed in 4% para-formaldehyde and subsequently paraffin-embedded. Tissues were dehydrated in xylene twice for five minutes each and washed using a gradient series of alcohol. The tissue sections for each sample were stained with hematoxylin–eosin (HE) and analyzed using a light microscope (Leica Microsystems, Wetzlar, Germany). The same methods were used for the other sample sections, which were blocked in 5% BSA for 2 h [34]. The tissues were washed with pre-cooled PBS twice for about 5 min. The sections were incubated with anti-SVV-VP1 mouse polyclonal antibody (1:5000) (made in our lab) at room temperature for 45 min. Subsequently, the sections were incubated with FITC-labeled goat anti-mouse antibody (1:5000) (Beyotime, Shanghai, China) at room temperature for 30 min. Intestinal dendritic cells were then used; CD11b mouse monoclonal antibody, FITC-labeled (1:2000), and DAPI were used to stain the nuclei. Then, all sections were washed with PBS for five minutes three times. Finally, all sections were observed under a light microscope, and fluorescence was observed with a fluorescence microscope in the dark (Leica Microsystems, Wetzlar, Germany).

### 4.6. The Production of Specific SVV Antibodies

All feces and sera were collected to examine the quantities of specific antibodies with the ELISA method. Each group randomly chose five piglets for sampling of feces and sera swabs. And three replications were performed for each sample. In our lab, the VP1 protein was obtained through prokaryotic expression. All serum and feces samples underwent double dilution. The secondary antibodies of horseradish peroxidase-conjugated (HRP) Rabbit Anti-pig IgG and Rabbit Anti-pig IgA/HRP (Solarbio, Beijing, China) were added, incubated for 1 h at RT, and then washed three times with PBST. The substrate o-phenylenediamine dihydrochloride (OPD) was used as the chromogen. All results were analyzed at 490 nm with an ELx800 microplate reader (BioTek, Winooski, VT, USA). The results for each group of plates were standardized using a panel of reference IgG and IgA negatives and positives. P/N ratios > 2 indicated positive antibodies.

### 4.7. Flow Cytometry

Peripheral blood lymphocytes were purified with blood lymphocyte isolation buffer, and then stained with PE-conjugated anti-mouse F4/80 (1:200, Biolegend, SanDiego, CA, USA) and FITC-conjugated anti-mouse CD11b (1:200, Biolegend, SanDiego, USA) for the gating macrophages [35]. PE-conjugated anti-mouse F4/80 together with APC-conjugated anti-human CD80 (1:200, Biolegend, SanDiego, CA, USA) was used to stain M1 phenotype macrophages, and the former with FITC-conjugated anti-mouse CD206 (1:200, Biolegend, SanDiego, CA, USA) was used to stain M2 phenotype macrophages. Cell surface antibody tests were performed by staining cells for 15 min at room temperature. CD206 antibody staining was performed using an intracellular staining kit, following the manufacturer’s protocol. Cells were analyzed on an LSRFortessa flow cytometer (BD Biosciences, Franklin Lakes, NJ, USA) using the Flowjo software (10.8.1, BD Biosciences, Franklin Lakes, NJ, USA).

### 4.8. Western Blot

A small amount of spleen tissue blockers was added to 400 μL of a single-detergent cracking solution, which was then homogenized within an Eppendorf tube in an ice bath. Then, the completely dissolved protein sample solution was added to a Coomassie bright blue solution, and the mixture was left to rest for a period of time, allowing the protein to fully mix with the solution. The absorbance of the mixture was analyzed at 595 nm using an ultraviolet spectrometer; the absorbance value was calculated by establishing the standard curve. Protein samples were injected into the SDS-PAGE gel, and SDS-PAGE was performed using methanol-activated PVDF membranes to transfer membranes in an ice bath for half an hour. The transferred membranes were blocked with 5%BSA overnight at 4 °C. The PVDF membranes were washed in PBST for 5 min, four times each, and then the first group of antibodies was added to the transferred membranes, including anti-SVV-VP1 antibody (1:5000), IFN-α antibody (1:2000), IFN-β antibody (1:2500), IFN-λ antibody (1:1000), TGF-β antibody (1:3000), IL-6 antibody (1:3000), and β-actin antibody (1:5000); all these antibodies were purchased from Company ABclonal, Inc. (Woburn, MA, USA). The second antibody group included HRP-labeled goat anti-mouse antibody (1:5000) and HRP-labeled goat anti-rabbit antibody (1:5000), added at room temperature for 1 h. Finally, ECL luminescent liquid was used to expose the PVDF membranes with a Western Blot Molecular Imaging System 300 (Azure Biosystems, Dublin, OH, USA). Finally, Image J/Fiji (1.6.0,NIH, Bethesda, MD, USA) was used to measure the gray values of the triplicate bands and the internal reference.

### 4.9. Statistical Analysis

Firstly, the weight changes in piglets following SVV infection, as well as the detection and analysis of IgG titers in different intestinal segments, were analyzed using a two-way ANOVA method. The detection and analysis of serum IgG levels and qRT-PCR Ct values in oral and fecal swabs were analyzed using ordinary one-way ANOVA. ImageJ/Fiji (NIH, Bethesda, MD, USA) software was employed to quantify the mean fluorescence intensity of immunofluorescence images and the gray values of Western blot bands. FlowJo (BD, Franklin Lakes, NJ, USA) software was utilized for the analysis of flow cytometry data. Statistical analysis was performed using ordinary one-way ANOVA for all aforementioned measurements. Secondly, the Kaplan–Meier method was used to estimate progression-free survival (PFS) and median survival time. Statistical significance was determined using the Log-rank (Mantel–Cox) test (*p* < 0.0001) and the Gehan–Breslow–Wilcoxon test (*p* < 0.001). Finally, all statistical analyses were conducted using GraphPad Prism (9.5.1, GraphPad software, SanDiego, CA, USA). ANOVA results were further analyzed using Tukey’s multiple comparisons test and the least significant difference (LSD) test. All samples and experiments were performed in triplicate, and all groups were treated as independent measurements to ensure sufficient statistical power. All observed effects were considered statistically significant at *p* < 0.05.

## 5. Conclusions

In terms of both cytokine expression profiles and the immunostimulatory phagocytic cell responses to SVV infection, Min piglets demonstrate superior performance compared to Landrace piglets. We observed a difference in the innate immunity of Min piglets, which is a key characteristic of breeding for disease resistance. Overall, this study is the first on the disease resistance of a Chinese native pig breed against SVV, taking the first step to obtain insights into the differences in the macrophages between Min piglets—as an indigenous breed from China—and Landrace piglets.

## Figures and Tables

**Figure 1 ijms-26-08746-f001:**
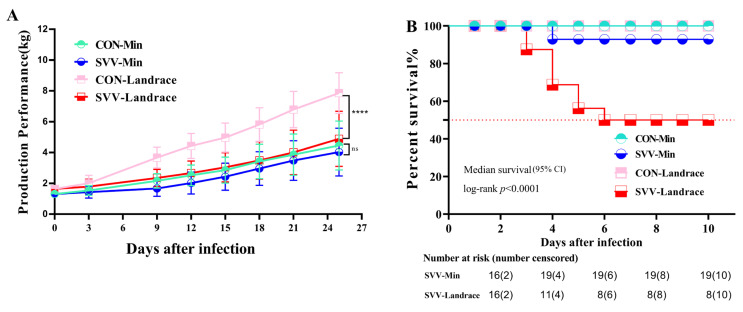
Min and Landrace pigs were challenged with SVV: production performance (**A**) and survival rate (**B**). (**A**) The Y-axis unit is kg, the X-axis represents the number of days after SVV infection. ns indicates no significant difference; **** indicates very significant difference (*p* < 0.0001). (**B**) The pigs were randomly divided into two groups, with 16 pigs in each group. For SVV-Landrace, n  = 16, median OS 6 day (95% CI). The Progression-Free Survival (PFS) and median survival time were estimated by Kaplan–Meier method, and the Log-rank (Mantel–Cox) test *p* < 0.0001. The table presents the number of surviving piglets over time, with the X-axis representing the corresponding days.

**Figure 2 ijms-26-08746-f002:**
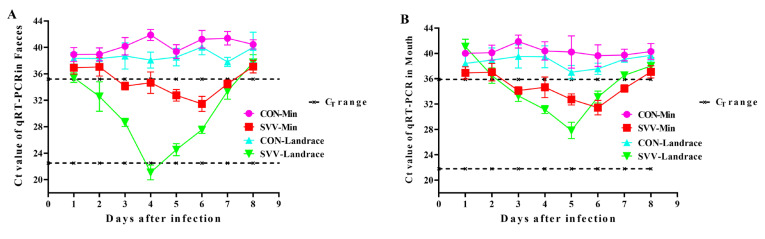
RT-qPCR was used to detect the titer of SVV in feces (**A**) and oral swabs (**B**) of Min pigs and Landrace pigs. The X-axis represents the number of days of SVV infection. The Y-axis represents the cycle threshold (Ct) value of RT-qPCR in feces (**A**) and mouth (**B**).

**Figure 3 ijms-26-08746-f003:**
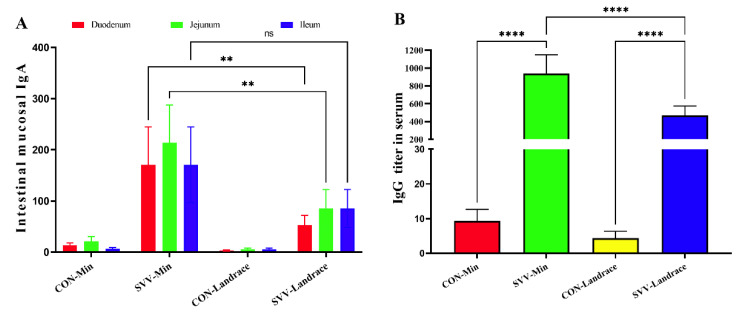
Enzyme-linked immunosorbent assay measures of specific SVV antibody titers of IgA and IgG in different intestinal segments and serum of Landrace and Min pigs. (**A**): Anti-SVV IgA titer in duodenum, jejunum, and ileum; (**B**): Anti-SVV IgG titer in serum. The Y-axis represents antibody titers measured in nanograms per milliliter (ng/mL). ns indicates no significant difference; ** indicates significant difference; **** indicates very significant difference (*p* < 0.0001).

**Figure 4 ijms-26-08746-f004:**
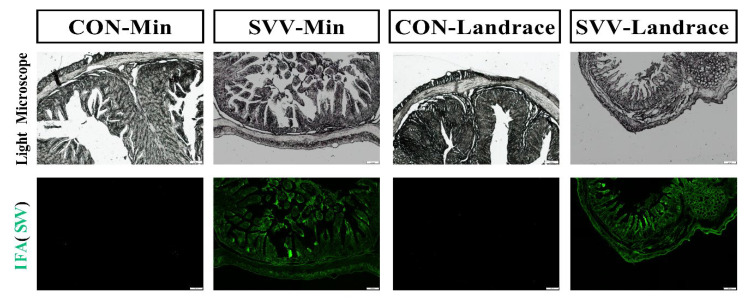
Pathology and immunofluorescence results indicating pathological changes and the quantity of the virus virion. Viral antigens were detected mainly in villous epithelial cells of the jejunum. White light microscopy and green light fluorescence results for the SVV antigen are presented in the first and second panels, respectively. The scale ruler of Figure 4 is 200 μm.

**Figure 5 ijms-26-08746-f005:**
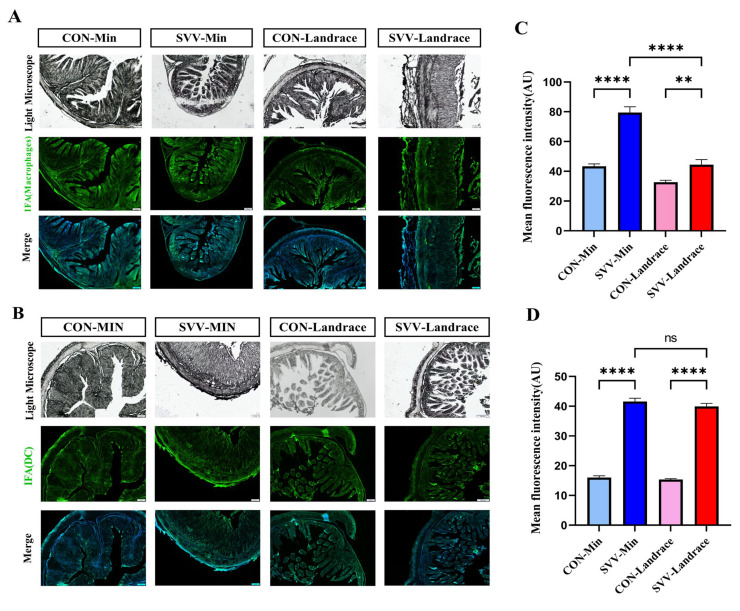
The immunofluorescence results of intestinal tissue indicating macrophage and dendritic cell activity. (**A**) F4/80 was used to label macrophages, and (**B**) CD11b was used to label dendritic cells. White light microscopy and green light fluorescence results for macrophages and dendritic cells are presented in the first and second panels, respectively. The third panel shows images obtained through the combined use of green fluorescence-labeled antibodies and DAPI nuclear staining. The scale ruler of Figure 5A,B is 200 μm. (**C**) Indicates the average fluorescence intensity of macrophages and the Y-axis is Arbitrary Units (AU). (**D**) Indicates the average fluorescence intensity of dendritic cells. ns indicates no significant difference; ** indicates significant difference; **** indicates very significant difference (*p* < 0.0001).

**Figure 6 ijms-26-08746-f006:**
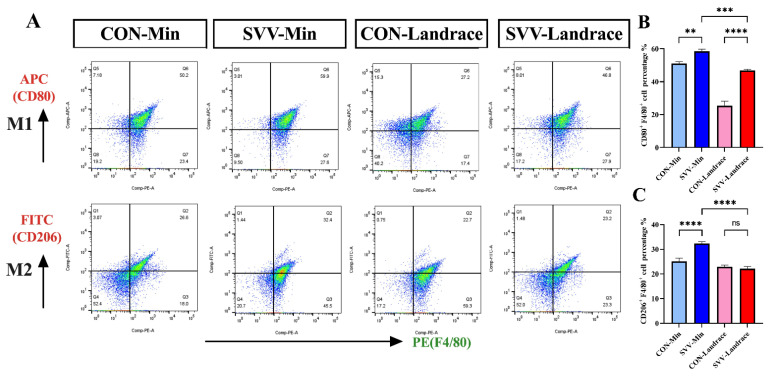
(**A**) Flow cytometric analysis results for the percentage of M1 and M2 macrophages for different pig breeds, showing the distribution of M1 macrophage subsets with dual fluorescence labeling of PE(F4/80) and APC(CD80) double fluorescence in the first panel and the distribution of M2 macrophage subsets labeled with PE(F4/80) and FITC(CD206) double fluorescence in the second panel. The abscissa represents the fluorescence intensity. (**B**) Indicates the percentage of CD80+ and F4/80+ cells. (**C**) Indicates the percentage of CD206+ and F4/80+ cells. ns indicates no significant difference; ** indicates significant difference; *** indicates very significant difference; **** indicates extremely significant difference (*p* < 0.0001).

**Figure 7 ijms-26-08746-f007:**
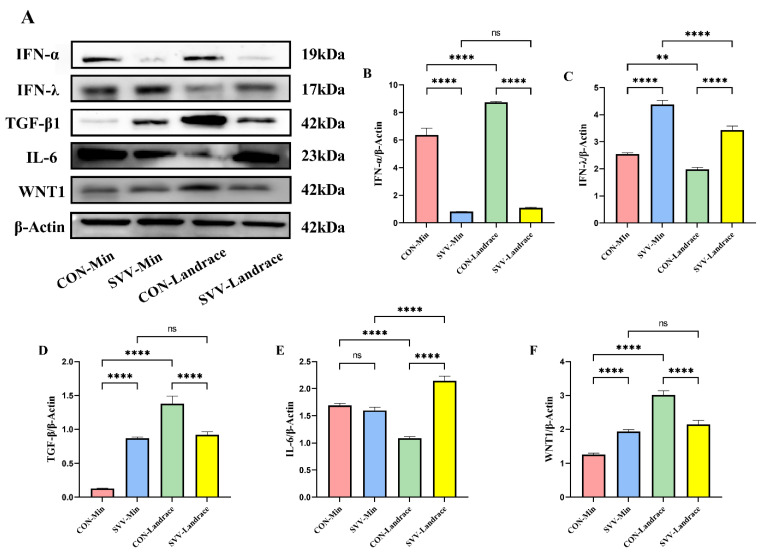
The expression of spleen tissue proteins was determined via Western blot (**A**) and quantitative analysis of protein expression (**B**–**F**). The protein expression levels of IFN-α (**B**), IFN-λ (**C**), TGF-β1 (**D**), IL-6 (**E**), and WNT1 (**F**) in the CON-Min, SVV-Min, CON-Landrace, and SVV-Landrace groups were detected using Western blot, with β-Actin used as an internal reference for correction. ns indicates no significant difference; ** indicates significant difference; **** indicates extremely significant difference (*p* < 0.0001).

## Data Availability

The authors confirm that the data supporting the findings of this study are available within the manuscript and tables.

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
