# Peer review of "Breed Varieties of Pigs for Disease Resistance and Susceptibility to Seneca Valley Virus Infection"

_ijms, 2025, doi:10.3390/ijms26178746_

Round 1

Reviewer 1 Report

Comments and Suggestions for Authors

This manuscript observed differences in susceptibility to SVV infection between two breeds of pigs and confirmed high resistance in Min pigs. This is an important study demonstrating disease resistance in Min pigs. However, the results and interpretation of fluorescence microscopy and flow cytometry analysis are insufficient and should be reconsidered. Therefore, this reviewer can not agree with the authors' discussions and conclusions. In addition, insufficient description of methods, results, or use of the terminology was found in a number of cases, and significant revisions are required throughout.

L43: Taxonomy should be included.

Results: Cite the figures in the text.

L83–84: Were the animals infected with different doses? That is not comparable. Were they inoculated with the same dose? 

L94–95: Were A and B the same experiment? The authors should add the infectious dose of virus in each experiment. In all of the following results, it is necessary to indicate whether the samples were derived from the same experimental infection.

Section 2.2: mRNA? viral genome? Which did the authors analyzed. Please specify it.

Figure 2: The viral titer should be indicated by copy number or relative values to a reference gene.

L111–113: Did the authors use the same animals in figure 1? Please clarify. Especially, viral dose (lethal dose?) should be clarified.

L132–134: Did the authors use the same animals in figure 1? Please clarify. Also, when did the authors collect the samples, and how many samples did the authors collect in each group?

L135: Spell out “ICA”.

L136: antiviral experiment: This description is very vague; please use other phrases. Does it mean the detection of viral antigens?

L142: Are they the results obtained by electron microscopy?

Section 2.4: The differences are hard to see. The interpretation should be reconsidered.

L148, Pathology: No mention of pathology is seen in the text.

Section 2.5: The results are unclear and do not appear to make a significant difference.

Section 2.6: No decisions can be made based on the results of one animal. What are the results of the other animals used in the experimental infection? The number of individuals should be increased and statistical analysis should be conducted. In addition, were macrophages in the intestines analyzed? Please clarify.

L198–199: Did the authors sort macrophages for the western blot analysis?

L201–202: No statistical tests to compare the difference between the two pig breeds were conducted, and therefore, the evidence is lacking.

L219: The abbreviation is already defined.

L228: What is SVA?

L256: period

L271–273: The data were not provided.

Section 4.3: The animal experiment in Figure 1A have to be described.

L303: Viral names should be spelled out. 

L307: 8.5 is superscript, 50 is subscript.

L311: d.p.c should be spelled out.

L317: Delete “ As mentioned previously”.

L317: Specify mRNA or viral genome as described above.

L328: Is the anti-PEDV-S an mAb specific for SVV antigens? Please add the explanation. Also, which viral proteins are recognized by this mAb?

L362: Which tissues should be described.

L370–374: Please clarify where the antibodies are purchased.

Comments on the Quality of English Language

This version is much more corrected than the previous version. However, minor errors are still included.

Author Response

Dear Reviewer 1,

We would like to extend our heartfelt thanks for your thoughtful suggestions regarding the manuscript. We truly value your insights and are happy to incorporate your revision recommendations. Below, you will find our line-by-line revisions made in response to your feedback. The updated sections are marked in yellow for your convenience.

  1. This manuscript observed differences in susceptibility to SVV infection between two breeds of pigs and confirmed high resistance in Min pigs. This is an important study demonstrating disease resistance in Min pigs. However, the results and interpretation of fluorescence microscopy and flow cytometry analysis are insufficient and should be reconsidered. Therefore, this reviewer cannot agree with the authors' discussions and conclusions. In addition, insufficient description of methods, results, or use of the terminology was found in a number of cases, and significant revisions are required throughout.

Thank you for your valuable comments. The Min pig is a local breed in northeast China, and there is limited research available on its disease resistance. We noticed that there were some areas in our descriptions of the immunofluorescence and flow cytometry sections that could be improved. We’ve already made the necessary revisions to ensure clarity and accuracy. We conducted qualitative analysis of macrophages from Min pigs and Landrace pigs using immunofluorescence, as well as quantitative analysis using flow cytometry.

  1. L43: Taxonomy should be included.

Thank you for your valuable comments. We have added a description of the viral taxonomic status of SVV at line 55.

  1. Results: Cite the figures in the text.

Thank you for your valuable comments. We have added all the cite of figure in the text, and we have corrected it in the revised manuscript.

  1. L83–84: Were the animals infected with different doses? That is not comparable. Were they inoculated with the same dose? 

Thank you for your suggestion, referring to a lot of literature, when the dose of virus infection is sufficient, the pathological damage and pathogenicity are mainly determined by the virus genotype. The genotype of Seneca virus is relatively conservative. And we have added the dose of virus for all the infecting group in the figure.

  1. L94–95: Were A and B the same experiment? The authors should add the infectious dose of virus in each experiment. In all of the following results, it is necessary to indicate whether the samples were derived from the same experimental infection.

Thank you for your comments. We have added the treatment conditions and virus dose in Figure 2. In Section 4.3 of Materials and Methods, we have improved the description of the animal experiment with virus infection.

  1. Section 2.2: mRNA? viral genome? Which did the authors analyzed. Please specify it.

Thank you for your comments. In this experiment, SVV viral load was determined mainly by detecting the copy number of SVV viral RNA in tissues by real-time fluorescence quantitative PCR. We have modified the title SVV RNA Copies number in two piglet breeds at L108.

Thank you for your valuable comments. In this experiment, the SVV viral load was primarily determined by measuring the copy number of SVV viral RNA in tissues using real-time fluorescent quantitative PCR. We have revised the title to “SVV RNA Copy Number in Two Piglet Breeds”at line 108.

  1. Figure 2: The viral titer should be indicated by copy number or relative values to a reference gene.

Thank you for your comments. We have noticed this problem and have modified the ordinate of Figure 2 to Ct value of qRT-PCR.

  1. L111–113: Did the authors use the same animals in figure 1? Please clarify. Especially, viral dose (lethal dose?) should be clarified.

L132–134: Did the authors use the same animals in figure 1? Please clarify. Also, when did the authors collect the samples, and how many samples did the authors collect in each group?

Thank you for your valuable comments. We have reused the same animals shown in Figure 1 and have updated the details, including the sample number and collection time, in Section 4.3 of the Materials and Methods. Additionally, we have marked the SVV dose used for infection in Figure 1.

  1. L135: Spell out “ICA”.

Thank you for your valuable comments. We sincerely apologize for the spelling mistake and have corrected it from ICA to IFA, and corrected it in the title.

  1. L136: antiviral experiment: This description is very vague; please use other phrases. Does it mean the detection of viral antigens?

Thank you for your comments. We found that the description was indeed inaccurate, and it has been revised to viral load of SVV antigens at L150

  1. L142: Are they the results obtained by electron microscopy?

Thank you for your valuable comments. We sincerely apologize for the incorrect expression. The correct term is “optical microscope,” and this has been revised at line 161.

  1. Section 2.4: The differences are hard to see. The interpretation should be reconsidered.

L148, Pathology: No mention of pathology is seen in the text.

Thank you for your valuable comments. We have added a detailed description of the pathological differences at lines 154–158.

  1. Section 2.5: The results are unclear and do not appear to make a significant difference.

Thank you for your valuable comments. The main conclusion of this section is that Min pigs macrophages exhibit stronger activity compared to Landrace pigs. Based on the immunofluorescence results, we observed that the immunofluorescence intensity in Min pig macrophages is higher than that in landrace pigs. These experiments were primarily used to characterize macrophage activity. And subsequently, flow cytometry (Section 2.6) emphasized to more accurately verify both macrophage numbers and cell typing.

  1. Section 2.6: No decisions can be made based on the results of one animal. What are the results of the other animals used in the experimental infection? The number of individuals should be increased and statistical analysis should be conducted. In addition, were macrophages in the intestines analyzed? Please clarify.

Thank you for your valuable comments. We have added details regarding the number of animals used in the Materials and Methods section. There were two animal breeds for the experiment. The SVV-induced diarrhea can cause damage to the intestinal mucosal immune system, and there is considerable individual variation. Since the immune response of each animal is likely determined by the peripheral immune system, splenic lymphocytes were analyzed using flow cytometry in this study.

  1. L198–199: Did the authors sort macrophages for the western blot analysis?

Thank you for your valuable comments. We have added detail information in the Western blot analysis at line 219.

  1. L201–202: No statistical tests to compare the difference between the two pig breeds were conducted, and therefore, the evidence is lacking.

Thank you for your valuable comments. We have added a comparison of the significant differences between Min pigs and Landrace pigs in Figure 7.

  1. L219: The abbreviation is already defined.

Thank you for your valuable comments. We have removed that expression as requested.

  1. L228: What is SVA?

Thank you for your valuable comments. As previously mentioned, SVV and SVA belong to the same virus, just with different names. For the accuracy and in respect for the cited reference, we have chosen to use the term SVA.

  1. L256: period

Thank you for your valuable comments. We are very sorry that we didn't understand your comments.

  1. L271–273: The data were not provided.

Thank you for your valuable comments. We sincerely apologize for the incorrect expression in lines 271–273. The correct term should be TGF-β instead of IL-10.

  1. Section 4.3: The animal experiment in Figure 1A have to be described.

Thank you for your valuable comments. We have added the description of the animal experiment from Figure 1 in Section 4.3 of the Materials and Methods.

  1. L303: Viral names should be spelled out. 

Thank you for your valuable comments. We have added the full spelling at lines 323–324.

  1. L307: 8.5 is superscript, 50 is subscript.

Thank you for your valuable comments. We have made the necessary modifications at the appropriate location.

  1. L311: d.p.c should be spelled out.

Thank you for your valuable advice. The abbreviation d.p.c. stands for "day post challenge".

  1. L317: Delete “As mentioned previously”.

Thank you for your valuable comments. We completely agree with your suggestion and have removed the phrase "As mentioned previously".

  1. L317: Specify mRNA or viral genome as described above.

Thank you for your valuable comments. We have revised the text at lines 343–344 to read: "The TaqMan real-time RT-PCR was used to detect the specific mRNA of SVV titers in pig organs, as well as the viral load, as described above."

  1. L328: Is the anti-PEDV-S an mAb specific for SVV antigens? Please add the explanation. Also, which viral proteins are recognized by this mAb?

Thank you for your comments. We are very sorry for our wrong expression. We have modified it to anti-SVV-VP1 mouse polyclonal antibody at L357.

  1. L362: Which tissues should be described.

Thank you for your advice. We used the spleen tissue in our WesternBlot experiment L390.

  1. L370–374: Please clarify where the antibodies are purchased

Thank you for your valuable comments. We have added the supplier information for each antibody at line 403.

Reviewer 2 Report

Comments and Suggestions for Authors

The manuscript entitled "Breed varieties of pigs for disease resistance and susceptibility to Seneca Valley virus infection” is interesting. However, before recommending publication of the paper, I would like to make a few points.

Abstract

L20 - There is a lack of methodological clarity – how many pigs were included? What age? What design? Please add brief information on this.

Introduction

The references in the Introduction section is not written in accordance with Instruction for authors / Template. Please check the whole manuscript and correct this. Also, the introduction could be more clearly structured, especially in the section on the relationship with SVV infection and the immunological differences between the breeds. Currently, the research background is fragmented and the rationale for the objectives is not sufficiently developed.

L42-L44 - PRRSV is introduced without any connection to the main topic (SVV). It is not clear to the reader why it is compared to another virus without explaining the immunological similarity.

L47-L50 – This sentence seems too long, I suggest to the Authors to rephrase this.

L50-L51 – Can you more pig breeds?

L56-L61 - The aims of the study are scattered over several overlapping sentences and use inconsistent wording such as “we hypothesised...”, “we aimed to evaluate...”, “we observed...”. Authors should state the research objective clearly and concisely

Results

L73 - The authors state significance, but no value is shown in this part of the manuscript. What is the p value? Neither in the text nor in the diagram (Figure 1A) is the SD/SEM shown. It is also not clear what “main difference” means.

L74 – “The birth and weaning weights of Min piglets were lower than those of Landrace piglets” Why? Is this a fundamental difference or a consequence of the infection? If they are already born smaller, then the infection does not explain the difference.

L75-L81 – Please add some numerical data, so that you can better understand the text and follow the results. It is important to show the differences and to add the p values.

L85 – “began to die by Day 3” rephrase as “first mortality was observed on Day 3 post-infection”

L87 - Mortality in the Landrace group is 50 %, in the Min group 10 %. This is a large difference, but the authors show no statistical analysis of survival (e.g. Kaplan-Meier + log-rank test).

L88 – “We chose Day 10 as the terminal point to access the significant difference between groups.”. Why? Please explain this.

L99 - Which is the target gene, it is not explained how the values were calculated (Ct in copies?)

L102 – Authors did not show no p-value, or how the difference was analysed. Where are the SD values?

L103-L107- This is similar sentence, please check and correct this.

L121-L124 - 200 doubling dilutions” – technically incorrect. The standard expression is “end-point titer of 1:200.” The same applies to 1:50 for Landrace.

L130 - There is no statistical confirmation of the stated differences (neither p, nor graph, nor error bars). The ELISA results should be presented with a well-defined cut-off value and a default value.

L156 – “three piglets” per group? Can you explain this small number of animals?

L161 – Authors stated that number of macrophages in Min piglets was higher than in Landrace piglets but we don’t have information on the number, how many?

L185 – “There was no significant difference between the two breeds” contradiction to the previous assertion. Here the Authors stated that there is “no” difference, but in the same paragraph they claim the opposite. This needs to be clarified.

L202 - What is the difference? No p values shown.

** In the entire manuscript, there is not a single table, p-value or specific numerical value to support the claim of statistically significant differences. The authors repeatedly state that the results are 'significant', 'higher' or 'lower', but these terms are not supported by a quantitative presentation of the data. This significantly reduces the credibility and scientific weight of the results.

Discussion

L217-L222 – This section is theoretical part not suitable for Discussion section, place this in the section of Introduction.

L225 – This is empty sentence, Authors should add their findings and results. Also, no citations of work from 2024/2025 are shown either – there is no review of more recent data.

L232 – “All groups were analyzed with SVV qPCR using mouth and fecal swabs in our 233 study [14].” Delete this, this is part of Material and methods.

L257 - The part about the role of phagocytes is important, but here it contradicts the previous statement. Authors claimed that Min pigs have stronger macrophage activity. It is necessary to clearly delineate what is evidence and what is speculation.

L260 – “breed characteristic of the genotype” rephrase as "a phenotype likely influenced by breed-specific genetic background”

L285 - The concluding part of the discussion is generalised. The claim that "these findings suggest that breeding pigs for immune-associated traits can benefit their disease resistance" is correct, but is not supported by experimental evidence from this work.

Materials and methods

L300-L302 - Important information is missing, e.g. how many pigs in total, how many in the experimental and control groups, the individual randomisation and whether the groups were kept under the same or separate conditions.

L320 - The qRT-PCR method is briefly described, but key elements are missing: which genes are targeted (only VP1?), which examples and probes are used, assay validation (sensitivity/specificity) and how the standard curve is generated

L338-L339 - What is the CV of the test? Were all measurements performed in duplicate/triplicate?

L244 - Is this typical of indigenous breeds? Is there evidence from other studies for similar phenotypic immune responses?

**The statistical analysis of the results is completely missing in the 'Materials and Methods' section, although the results repeatedly show significant differences. The Authors must clearly state which tests were used, how the significance limits were defined and provide information on the sample size and data variability. Otherwise, the conclusions presented in the paper remain untested and methodologically questionable.

** Author Contributions – This is not written in accordance with Instruction for authors.

**References – must be corrected and written in accordance with Template or Instruction for authors.

Author Response

Dear Reviewer 2,

We would like to extend our heartfelt thanks for your suggestions regarding our manuscript. We truly pay attention to your insights, and are happy to incorporate your revision. Below, you will find our line-by-line revisions made in response to your feedback. The updated sections are marked in yellow for your convenience.

The manuscript entitled "Breed varieties of pigs for disease resistance and susceptibility to Seneca Valley virus infection” is interesting. However, before recommending publication of the paper, I would like to make a few points.

Abstract

  1. L20 - There is a lack of methodological clarity – how many pigs were included? What age? What design? Please add brief information on this. 

Thank you very much for your valuable comments. We have made a minor correction at line L19. In addition, we have included the pig’s number and age of piglets in Section 4.3 (Materials and Methods) to provide more clarity.

Introduction

  1. The references in the Introduction section is not written in accordance with Instruction for authors / Template. Please check the whole manuscript and correct this. Also, the introduction could be more clearly structured, especially in the section on the relationship with SVV infection and the immunological differences between the breeds. Currently, the research background is fragmented and the rationale for the objectives is not sufficiently developed.

Thank you very much for your thoughtful comments. We have carefully revised the references in the introduction section following the journal’s template. In addition, we have restructured the logic of introduction part, and improved the research background section of the introduction, in order to provide a clearer explanation of the relationship between SVV infection and the immunological differences among different breeds.

  1. L42-L44 - PRRSV is introduced without any connection to the main topic (SVV). It is not clear to the reader why it is compared to another virus without explaining the immunological similarity.

Thank you for your suggestion. The introduction of PRRSV was to illustrate the disease resistance characteristics of local pig breeds in China, thereby leading to the introduction of Min pig's disease resistance. And we have also added another reference to focus on the relationship the local pig breeds and disease-resistance. Certainly, we have added the research of Min pig in the disease-resistance.

  1. L47-L50 – This sentence seems too long, I suggest to the Authors to rephrase this.

Thank you for your thoughtful comment. We have revised the sentence in lines 61–62 to enhance its clarity and conciseness. And we have also added relative reference about Min pig on the disease-resistance.

  1. L50-L51 – Can you more pig breeds?

Thank you for your valuable feedback. We have identified that the description in lines 50–51 contained inaccuracies and have therefore removed this section.

  1. L56-L61 - The aims of the study are scattered over several overlapping sentences and use inconsistent wording such as “we hypothesised...”, “we aimed to evaluate...”, “we observed...”. Authors should state the research objective clearly and concisely

Thank you for your thoughtful comments. As you pointed out, the research objectives in this section were indeed somewhat scattered. We have revised the content in lines 70–75 to make it more focused and to more clearly convey the research purposes.

Results

  1. L73 - The authors state significance, but no value is shown in this part of the manuscript. What is the p value? Neither in the text nor in the diagram (Figure 1A) is the SD/SEM shown. It is also not clear what “main difference” means.

Thank you for your valuable comments. We have included the P-values and corresponding significance analysis in Figure 1 to enhance the clarity of the results.

  1. L74 – “The birth and weaning weights of Min piglets were lower than those of Landrace piglets” Why? Is this a fundamental difference or a consequence of the infection? If they are already born smaller, then the infection does not explain the difference.

Thank you for your valuable input. It is true that the birth weight and weaning weight of Min pigs are generally lower than those of Landrace pigs, which reflects the genetic characteristics of Min pigs. Our objective is to compare the extent of weight loss in both Min and Landrace pigs with SVV infection relative to their respective control groups, rather than to directly compare their intra-group differences weight values.

  1. L75-L81 – Please add some numerical data, so that you can better understand the text and follow the results. It is important to show the differences and to add the p values.

Thank you for your helpful suggestion. We sincerely apologize for the lack of clarity in this section. To improve the presentation, we have included the P values and corresponding significance analysis in Figure 1.

  1. L85 – “began to die by Day 3” rephrase as “first mortality was observed on Day 3 post-infection”

Thank you for your thoughtful comment. We sincerely apologize for the lack of clarity in the original description. To improve the wording, we have revised it to “First mortality was observed on Day 3 post-infection”at line 91.

  1. L87 - Mortality in the Landrace group is 50 %, in the Min group 10 %. This is a large difference, but the authors show no statistical analysis of survival (e.g. Kaplan-Meier + log-rank test).

Thank you for your valuable comments. We have performed the Kaplan-Meier analysis in combination with the log-rank test. We sincerely apologize for not including this information in the previous version of the manuscript. To enhance the completeness of the statistical analysis, we have added the relevant results in Figure 1B. And we have also added the number of surviving piglets over time.

  1. L88 – “We chose Day 10 as the terminal point to access the significant difference between groups.”. Why? Please explain this.

Thank you for your valuable feedback. From the perspective of veterinary virology, the duration of Seneca Valley Virus (SVV) infection is typically short, lasting approximately 3 to 5 days. And there was no death case at the sixth day. Therefore, we have selected 10 days as the observational endpoint to better capture the differences in survival rates among various pig breeds following SVV infection.

  1. L99 - Which is the target gene, it is not explained how the values were calculated (Ct in copies?)

Thank you for your comments. The details of this part can be found in Materials and Methods 4.3 L341-348. Ct represents the cycle number of real-time fluorescent quantitative PCR. The lower the Ct value, the greater the amount of viral RNA. We have conducted preliminary research on Seneca virus and added the reference for our recent research on the application of this method to the Materials and Methods section.

  1. L102 – Authors did not show no p-value, or how the difference was analysed. Where are the SD values?

Thank you for your valuable feedback. The area between the two dotted lines in Figure 2 represents the effective range of viral load. A Ct value that is excessively high and exceeds the upper dotted line indicates that the viral content in the tissue is too low, did not to confirm an infection from a veterinary clinical testing of SVV infection. Conversely, a Ct value that is excessively low and falls below the lower dotted line suggests that the viral content may be too high, which could indicate a potential false positive result or errors during tissue sample processing.

  1. L103-L107- This is similar sentence, please check and correct this.

Thank you for your valuable feedback. We sincerely apologize for the repetition in the text. To improve clarity and readability, we have revised the duplicated sentences in lines 115–121.

  1. L121-L124 - 200 doubling dilutions” – technically incorrect. The standard expression is “end-point titer of 1:200.” The same applies to 1:50 for Landrace.

We appreciate your valuable feedback regarding the technical terminology. You are absolutely correct — the correcting expression should be “end-point titer of 1:200” rather than “200 doubling dilutions.” The same correction has been applied to the 1:50 value for Land.

  1. L130 - There is no statistical confirmation of the stated differences (neither p, nor graph, nor error bars). The ELISA results should be presented with a well-defined cut-off value and a default value.

Thank you for your valuable suggestion. We have included the significance analysis in Figure 3 and clearly labeled the P values.

  1. L156 – “three piglets” per group? Can you explain this small number of animals?

Thank you for your valuable feedback. We selected three replication piglets for the experiment. Furthermore, from a veterinary perspective, the number of three replications was enough to observe macrophages with immunofluorescence.

  1. L161 – Authors stated that number of macrophages in Min piglets was higher than in Landrace piglets but we don’t have information on the number, how many?

Thank you for your valuable feedback. In the immunofluorescence section, we conducted an experiment to observe the general presence and distribution of macrophages. Regarding the quantitative assessment of macrophage number, we have presented the results in the subsequent flow cytometry section of this study. The flow cytometry results were much more accuracy.

  1. L185 – “There was no significant difference between the two breeds” contradiction to the previous assertion. Here the Authors stated that there is “no” difference, but in the same paragraph they claim the opposite. This needs to be clarified.

Thank you for your valuable comment. We sincerely apologize for the typographical error and have revised the sentence to: "There was significant difference between Min piglets and Landrace piglets in the M1 macrophage polarization," on lines 203–204.

  1. L202 - What is the difference? No p values shown.

Thank you for your valuable suggestion. We have included the inter-group comparison between Min pigs and Landrace pigs in Figure 7, along with the significance analysis and corresponding P values.

  1. ** In the entire manuscript, there is not a single table, p-value or specific numerical value to support the claim of statistically significant differences. The authors repeatedly state that the results are 'significant', 'higher' or 'lower', but these terms are not supported by a quantitative presentation of the data. This significantly reduces the credibility and scientific weight of the results.

Thank you for your valuable comments. We sincerely apologize for not clearly indicating the differences and P values in the previous version of the manuscript. We have now added the P values and explanations of the significance analysis in the Figure and its corresponding figure legend.

Discussion

  1. L217-L222 – This section is theoretical part not suitable for Discussion section, place this in the section of Introduction.

Thank you for your valuable comments. We have corrected the relevant sentences into the Introduction section and removed the remaining inappropriate expressions.

  1. L225 – This is empty sentence, Authors should add their findings and results. Also, no citations of work from 2024/2025 are shown either – there is no review of more recent data.

Thank you for your valuable comments. We agreed that there was problem in our previous expression, and have now revised the relevant section. The cited reference has the relationship between neonatal diarrhea, pathological damage and our study results, which was better align with the focus of this study.

  1. L232 – “All groups were analyzed with SVV qPCR using mouth and fecal swabs in our study [14].” Delete this, this is part of Material and methods.

Thank you for your valuable suggestion. We have deleted this section to the Materials and Methods part for better clarity and organization.

  1. L257 - The part about the role of phagocytes is important, but here it contradicts the previous statement. Authors claimed that Min pigs have stronger macrophage activity. It is necessary to clearly delineate what is evidence and what is speculation.

Thank you for your valuable comments. The previous immunofluorescence and flow cytometry results indicated that the macrophage activity in Min pigs was higher compared to that in Landrace pigs.

  1. L260 – “breed characteristic of the genotype” rephrase as "a phenotype likely influenced by breed-specific genetic background”

Thank you for your valuable comments. We have revised the text to "a phenotype likely influenced by breed-specific genetic background" on line 277–278.

  1. L285 - The concluding part of the discussion is generalised. The claim that "these findings suggest that breeding pigs for immune-associated traits can benefit their disease resistance" is correct, but is not supported by experimental evidence from this work.

Thank you for your valuable comments. We have revised lines 302–304 to clarify that the enhanced macrophage activity and IFN-λ expression in Min pigs may contribute to their stronger disease resistance.

Materials and methods

  1. L300-L302 - Important information is missing, e.g. how many pigs in total, how many in the experimental and control groups, the individual randomisation and whether the groups were kept under the same or separate conditions.

Thank you for your valuable comments. We sincerely apologize for the omission of this information. We have now added the detailed descriptions of both the experimental group and the control group in Animal Experiment Section 4.3 in the Materials and Methods.

  1. L320 - The qRT-PCR method is briefly described, but key elements are missing: which genes are targeted (only VP1?), which examples and probes are used, assay validation (sensitivity/specificity) and how the standard curve is generated.

Thank you for your valuable comments. We have added a section detailing the methods used in our previous study, including qRT-PCR and the construction of its standard curve, on lines 346–348.

  1. L338-L339 - What is the CV of the test? Were all measurements performed in duplicate/triplicate?

Thank you for your valuable comments. We have added the following description in lines 367–368: "Each group randomly selected five piglets for fecal and serum swab sampling, and each sample was tested in triplicate."

  1. L244 - Is this typical of indigenous breeds? Is there evidence from other studies for similar phenotypic immune responses?

Thank you for your valuable comments. The Min pig is a unique breed native to northeast China, and the article also cites studies conducted by other research teams on its disease resistance.

  1. **The statistical analysis of the results is completely missing in the 'Materials and Methods' section, although the results repeatedly show significant differences. The Authors must clearly state which tests were used, how the significance limits were defined and provide information on the sample size and data variability. Otherwise, the conclusions presented in the paper remain untested and methodologically questionable. 

Thank you for your valuable comments. We sincerely apologize for the omission of the statistical analysis section in the previous version of the manuscript. We have now added this section of 4.9 (Statistical analysis) in the last part (Lines 411–425) of the Materials and Methods.

  1. ** Author Contributions – This is not written in accordance with Instruction for authors.

Thank you for your comments, we have revised it in accordance with the author contribution format required by the journal.

  1. **References – must be corrected and written in accordance with Template or Instruction for authors.

Thank you for your valuable comments. We have revised the reference format in accordance with the journal’s requirements.

Round 2

Reviewer 1 Report

Comments and Suggestions for Authors

Major comments

This manuscript has been improved in several points. Differences were also observed in survival rates after virus inoculation, antibody production, and cytokine expression, suggesting disease resistance. However, the presentation of important data to draw the conclusion that macrophage activity is high is still insufficient in the following points.

Section 2.5: There is still no clear difference. It is necessary to show that there are obvious characteristics common to multiple individuals, rather than presenting data from only one sample. The authors claim that the fluorescence intensity is high, but it is necessary to show clearly objective data.

Section 2.6 The results of flow cytometric analysis cannot be considered objective, unless they are analyzed using samples from multiple animals and the averages were compared. Therefore, the current data lacks reliability.

Other comments

L43: PRRSV should be spelled out.

L88–89: The text should be reconsidered.

Figure 1: Although the observation periods between A and B differ, were experiments A and B conducted using the same animals? Or were they conducted using different animals? The explanation remains unclear.

Section 2.2: The text states that mRNA was detected. It may be difficult to distinguish mRNA from viral genome. If so, this should be noted in the text.

L124: This is not copy number.

L149–150: This is an experiment to detect viral antigens.

L201–202: Which data were statistically analyzed to claim a significant difference?  

L218–219: The reviewers' comments regarding the materials used in the Western blot have not been addressed. It is necessary to clarify why the spleen was used in this analysis. 

L241: SVA is not defined, and the reviewer’s comments have not been addressed.

L269: Cite appropriate references.  

L326–329: The sentences should be reconsidered.  

L331–333: The sentence should be reconsidered.  

L333: (d.p.c) is unnecessary.  

L340–341: The sentence should be reconsidered.  

L399: Insert a space before “all”.  

Author Response

Dear Reviewer 1,

We would like to extend our heartfelt thanks for your thoughtful suggestions regarding the manuscript. We truly value your insights and are happy to incorporate your revision recommendations. Below, you will find our line-by-line revisions made in response to your feedback. The updated sections are marked in yellow for your convenience.

Comments and Suggestions for Authors

Major comments

This manuscript has been improved in several points. Differences were also observed in survival rates after virus inoculation, antibody production, and cytokine expression, suggesting disease resistance. However, the presentation of important data to draw the conclusion that macrophage activity is high is still insufficient in the following points.

Section 2.5: There is still no clear difference. It is necessary to show that there are obvious characteristics common to multiple individuals, rather than presenting data from only one sample. The authors claim that the fluorescence intensity is high, but it is necessary to show clearly objective data.

Thank you for your comments. We have test three replications for every group. In the new manuscript, we added the statistical analysis of bar graph for our flow cytometric data. If necessary, we could provide the original picture for flow cytometric analysis. Certainly, we added a new statistical analysis for pathology and immunofluorescence results in section 2.5.

Section 2.6 The results of flow cytometric analysis cannot be considered objective, unless they are analyzed using samples from multiple animals and the averages were compared. Therefore, the current data lacks reliability.

Thank you for your comments. We have test three replications for every group. In the new manuscript, we added the statistical analysis for our flow cytometric data. If necessary, we could provide the original picture for flow cytometric analysis.

Other comments

L43: PRRSV should be spelled out.

Thank you for your comments. We have added the full spelling of PRRSV at line 43.

L88–89: The text should be reconsidered.

Thank you for your valuable comments. We agree that there were problem with the description in lines 88–89, and we have made the necessary revisions in line 89. The change part have been marked with the red background.

Figure 1: Although the observation periods between A and B differ, were experiments A and B conducted using the same animals? Or were they conducted using different animals? The explanation remains unclear.

Thank you for your comments. We used the same piglets in Figure 1A for monitoring performance as those in Figure 1B, and conducted to monitor weight for survived piglets with challenging the SVV. This description has been revised at line 79.

Section 2.2: The text states that mRNA was detected. It may be difficult to distinguish mRNA from viral genome. If so, this should be noted in the text.

Thank you for your comment. The genome of the SVV virus is a single-stranded, positive-RNA molecule that can function as mRNA. In our experiment, cDNA was obtained by reverse transcription of viral RNA to cDNA, which was used as a template for real-time PCR amplification. We have also added some details about this method in the section 4.4 L348-L351.

L124: This is not copy number.

Thank you for your comments. It was our mistakes. We have revised this description at line 124.

L149–150: This is an experiment to detect viral antigens.

Thank you for your comments. We identified the problem with the description in this sentence, and it has been deleted in the new manuscript.

L201–202: Which data were statistically analyzed to claim a significant difference?

Thank you for your valuable comments. We have revised Figure 6 and added bar charts along with statistical analysis.

L218–219: The reviewers' comments regarding the materials used in the Western blot have not been addressed. It is necessary to clarify why the spleen was used in this analysis. 

Thank you for your suggestion. The spleen was one of the most important peripheral immune organs. From previous results we could find that most of results focus on the innate immunity, and we supposed that the possible molecular mechanism about immune factor were on the immune system. We did not choose the intestinal organs. Because the intestine was close on the mucosal immune system.

L241: SVA is not defined, and the reviewer’s comments have not been addressed.

Thank you for your comments. SVA is defined at line 55 and refers to the SVV virus.

SVV and SVA were the same virus, just with different names. Recent years, we have uniformed naming SVV. In the reference, the author used the SVA in the study. We have chosen to use the term SVV for easy to understand for readers.

L269: Cite appropriate references.  

Thank you for your comments. We have added relevant references at line 269.

L326–329: The sentences should be reconsidered.  

Thank you for your comments. We have revised the text in lines 328–332. We have corrected relative part in the expression, and the correcting part have been marked with red background,

L331–333: The sentence should be reconsidered.  

Thank you for your comments. We have revised the corresponding text in lines 335–337.

L333: (d.p.c) is unnecessary.  

Thank you for your comments. We found that the abbreviation was indeed unnecessary and have removed it.

L340–341: The sentence should be reconsidered.  

Thank you for your comments. We have revised the corresponding text in lines 345–346.

L399: Insert a space before “all”.  

Thank you for your comments. We have added appropriate spacing at line 399.

Reviewer 2 Report

Comments and Suggestions for Authors

Dear Authors,

Thank You for the revised version of Your manuscript.

Author Response

Dear Reviewer 2, 

Thank you very much for your thoughtful feedback, your generous time, and the valuable comments you provided throughout the review process. We truly appreciate your support and professionalism. 

With your careful review and insightful suggestions, the overall quality of our paper has been greatly enhanced.

Authors